# Improving Oral Presentation Skills for Radiology Residents through Clinical Session Meetings in the Virtual World Second Life

**DOI:** 10.3390/ijerph20064738

**Published:** 2023-03-08

**Authors:** Alberto Pino-Postigo, Dolores Domínguez-Pinos, Rocío Lorenzo-Alvarez, José Pavía-Molina, Miguel J. Ruiz-Gómez, Francisco Sendra-Portero

**Affiliations:** 1Department of Radiology, University Hospital Virgen de la Victoria, 29010 Málaga, Spain; apinomlg@gmail.com (A.P.-P.); loly1977@hotmail.com (D.D.-P.); 2Department of Radiology and Physical Medicine, Faculty of Medicine, University of Malaga, 29071 Málaga, Spain; mjrg@uma.es; 3Department of Emergency and Critical Care, Hospital de la Axarquía, 29700 Vélez Málaga, Spain; rociolorenzoalvarez@gmail.com; 4Department of Pharmacology, Faculty of Medicine, University of Malaga, 290071 Málaga, Spain; pavia@uma.es

**Keywords:** radiology education, virtual worlds, residents, oral presentation, clinical sessions

## Abstract

Background: The objective of this study was to conduct a clinical session meeting in the virtual world of Second Life to improve the oral presentation skills of radiology residents and to assess the perception of the attendees. Methods: A clinical session meeting (10 two-hour sessions over four weeks), where participants presented their own clinical sessions, followed by a turn of interventions by the attendees, was designed and carried out. Attendees were asked to complete an evaluation questionnaire. Descriptive statistics were performed. Results: Twenty-eight radiology residents attended the meeting, and 23 (81.2%) completed the evaluation questionnaire; 95.7–100% of them agreed that the virtual environment was attractive and suitable for holding the meeting and that the content was appropriate for their training as residents. They rated with ≥8.9 points (from 1 to 10) different aspects of the experience, highlighting the role of teachers (9.7 ± 0.6) and the usefulness of their training (9.4 ± 0.9). Conclusions: Second Life can be used effectively to train oral communication skills in public, in an environment perceived as attractive and suitable for learning, through an experience described by the attendees as interesting and useful, highlighting the advantages of social contact with their peers.

## 1. Introduction

### 1.1. Medical Education in Oral Communication Skills in Public

Public speaking is a learnable skill that requires communication skills to convey information and keep an audience interested. Throughout their training and practice, physicians are often asked to speak publicly on different topics, to diverse audiences, and in varied settings, from large academic meetings to informal briefings and impromptu discussions, including daily sessions in hospital departments. Many of the essential communication skills are the same and can be learned, practiced, and even honed [1].

Effective public speaking skills should begin to be acquired in college, as they indicate creativity, critical thinking, leadership, self-confidence, and professionalism, thereby helping students advance their careers and prepare for the job market [2]. Although there is interest in including the teaching of communication skills in medical schools, this interest often only contemplates doctor-patient communication [3], leaving public communication skills aside. Unfortunately, medical school, residency, fellowship, and other graduate training in medicine and science offer little or no specific instruction or formal guidance about presentation skills [1]. The residency is a good learning period to fill this gap and develop training activities in communication skills, taking advantage of online technology, and developing educational strategies beyond the needs imposed by the COVID-19 pandemic [4,5,6].

### 1.2. Clinical Sessions during the Radiology Residency

General competencies during radiology residency training include six categories [7]: patient care, medical knowledge, interpersonal relationships and communication, professionalism, practice-based learning and improvement, and systems-based learning. Currently, interpersonal and communication skills are increasingly important for radiologists [8]. Communication skills, although often neglected in medical schools and in daily hospital practice, constitute an important transversal competence in specialized medical training, regardless of the medical specialty. They include the preparation of appropriate written reports; oral communication, with the patient and/or his/her relatives, between colleagues and health professionals from the same department and from the rest of the hospital; and speaking in public in professional forums and clinical sessions.

Clinical sessions are a common practice in all hospital departments, which usually take place first thing in the morning. An adequate number and quality of clinical sessions are necessary for the accreditation of a resident training department. Clinical sessions in radiology departments can be grouped into five different types [9]: case-reading-based, thematic exposition, bibliographic review, discussion of clinical protocols, and department-related issues. In these sessions, radiological reading and diagnostic reasoning are improved, and the professional language of the residents is modeled to a great extent. Additionally, these sessions provide an important source of continuing education for radiologists.

Residents used to participate in clinical sessions by answering questions posed by radiologists, but often they prepare and present the daily clinical session themselves as part of their training [10]. To carry out this type of session, good public communication skills and the use of appropriate, correct, and precise language are required, as well as the preparation of support material, such as PowerPoint presentations or similar, that complement the presentation in a balanced way. Residents often learn through clinical experience throughout their training period, but specific training in these types of skills would be appropriate, both in medical school and during residency.

### 1.3. Multi-User Virtual Worlds

Current communication technologies allow remote learning of radiology through synchronous sessions. This has taken off since 2020 due to the COVID-19 pandemic, with abundant digital learning resources for students offered by individual educators, institutions and professional societies, providing alternatives to face-to-face teaching [4]. The use of remote learning as a daily clinical teaching tool instead of real time face-to-face clinical sessions has been addressed with technologies based on two-dimensional (2D) communication tools such as Zoom [5,6]. Multi-user virtual worlds are another current communication technology that consists of generating three-dimensional (3D) scenes on the computer screen, where users interact through a representation of themselves called an avatar [11]. They are multi-user platforms that support a set of human activities that improve ways of learning, allowing file sharing, immersive conferences, and simulations [12]. The concept of multi-user virtual worlds, along with others such as virtual reality, mirror worlds, and augmented reality, integrates the concept of the metaverse, a virtual reality beyond reality, referring to the digitized earth as a new world expressed through digital media and global connection via the internet [13].

Virtual worlds are multi-user platforms that allow immersive online meetings with a high sense of presence in a setting that can be like the real world, such as classrooms or meeting rooms, resulting in a less cold and boring environment than the screen of 2D webinar platforms. Virtual worlds have several characteristics, including [14]:A persistent environment 24/7.A shared space that allows multiple users to participate simultaneously.Virtual incarnation in the form of an avatar (a customizable three-dimensional representation of the user).Interactions between users and objects in the 3D environment.Immediate action so that interactions occur in real time.Similarities to the real world, such as topography, motion, and physics, provide the illusion of being there.

Second Life, launched in 2003 by Linden Lab (San Francisco, CA, USA), is one of the most widely used virtual worlds in higher education [11,14,15,16,17]. Users can interact with each other, communicating by voice or written chat, which makes it possible to carry out synchronous educational experiences in health sciences [18] and specifically in radiology [19]. The radiology educational activities carried out synchronously in Second Life, such as courses, workshops or seminars, have a great acceptance and educational recognition among both undergraduate students and medical graduates [20] and have demonstrated an impact on learning equal to that of the classroom in the real world, when the same conditions and educational contents are reproduced [21].

Practice in giving speeches and communicating in front of an audience has been shown to reduce communication apprehension. Computer-mediated communication can be a solution for those who are shy or have difficulty speaking in front of an audience. In other words, giving speeches in a virtual reality context can be effective in reducing communication apprehension [22]. Multi-user virtual worlds such as Second Life seem to be useful for people who are particularly reluctant to speak in public [23]. The multimodal 3D resources of Second Life have been shown to provide visual and linguistic support that facilitates foreign language learning for adults [24] and reduces anxiety levels in oral interaction [25]. In a recent study, Second Life has been shown to enable biomedical engineering students to present scientific content to their peers, receiving hands-on training in data collection, organization, and presentation tasks, with the benefits of remote access, collaborative work, and social interaction [26].

Radiology residents, like those of other specialties, would benefit from specific courses that allow them to train their communication skills in public. Multi-user virtual world platforms such as Second Life offer interesting, cost-effective learning resources to explore. The authors hypothesize that these resources can be used for training radiology residents in public presentation skills in virtual settings such as daily hospital clinical sessions. The objective of this study was to conduct a clinical sessions meeting in the virtual world of Second Life to improve the oral presentation skills of radiology residents and to assess the perception of the attendees.

## 2. Materials and Methods

A meeting of clinical sessions for radiology residents was designed, consisting of 10 two-hour sessions over four weeks. The meeting was held in the virtual world of Second Life, specifically in a virtual place called Medical Master Island [27]. It started with several talks on how to make good oral presentations. Subsequently, each participant had to present a clinical session to the attendees, followed by a round of interventions from the attending peers, professors, and guest visitors. After the meeting ended, the participants were asked to complete a questionnaire to evaluate the experience.

### 2.1. The Virtual Location

The Medical Master Island is a virtual island located on a 256 × 256 m square piece of virtual land. The island was designed to resemble a university campus, with three main buildings: the Medical Master Conference Center, the Undergraduate Building, and the Postgraduate Building (Figure 1). The synchronous sessions of this study were carried out in a semi-basement of the Postgraduate Building. There, the clinical sessions took place around a U-shaped table. The presenters had a wall screen for the slide presentation. Next to the clinical session room, an exhibition room was set up so that the participants could review the presentations and videos of the sessions after their presentation (Figure 2).

### 2.2. The Clinical Session Meetings

#### 2.2.1. Curriculum Design

The didactic objective of the clinical session meetings was to improve communication skills in scientific settings in relation to the preparation of supporting slide shows and oral presentations. The specific objectives were: (1) to develop the learning of non-interpretive oral communication skills by radiology residents in an immersive environment; (2) to evaluate the impact of the self-evaluation and peer evaluation of the oral presentations of the participants as a practical element of improvement, and (3) to integrate all this in a participative environment supported mainly by voice chat.

#### 2.2.2. Learning Design

The meetings consisted of 10 sessions of 2 h over 4 weeks, which are summarized in Table 1. The first day was dedicated to welcoming the participants and visiting the island, an introduction to Second Life and a presentation of the meeting. During this first session, specific tasks were carried out with the Second Life viewer interface, training the avatar movement controls, operating the microphone, audio reception, and camera handling (avatar vision). The next three days were spent giving lectures on what the clinical sessions consist of, based on the paper by Del Cura Rodríguez [9], how to speak in public [28,29,30] and how to prepare proper PowerPoint presentations [31,32,33,34]. In these sessions, attendees were trained to follow an oral presentation in Second Life, encouraging their intervention by raising doubts or questions about the topic of the day. The next five days were devoted to presenting clinical sessions. Groups and turns were organized so that each participant presented their own clinical session. Participants were required to submit their PowerPoint presentations in advance to the organizers. The presentations were turned into a web page with forward and back buttons so they could be displayed in Second Life. On the assigned day, each attendee had 30 min to present the clinical session to the audience. After the presentation, there was a round of interventions by the attendees, including professors and guests, in which each one expressed their opinion on the content of the session, the slide presentation format, and the oral presentation. The last day was dedicated to a closing session, in which the attendees presented their personal impressions, and the final conclusions of the meeting were raised.

After each day, the presentations remained on display in the exhibition room adjacent to the clinical session room. In addition, the daily sessions were recorded with Camtasia^®^, making streaming on-demand video available to attendees throughout the experience. This allowed participants who could not attend any of the days (e.g., they were on call) to review the content afterward. When the presentations of the day concluded before two hours, the final minutes were dedicated to visiting, with those attendees who wanted to, other Second Life places of an educational, artistic, cultural, or leisurely nature. This contributed to fostering interaction among the group and provided participants with a more precise knowledge of the versatility of this multi-user virtual world.

Between November 2012 and October 2014, four clinical session meetings for radiology residents were held. Spanish radiology residents were previously invited via email and social networks, with the collaboration of the Spanish Association of Medical Radiology (SERAM), its regional affiliates and the undergraduate and postgraduate training section (FORA). With the invitation, they were informed about the program and content of the meeting and that registration was free for SERAM members. The only requirements to participate were being a radiology resident, creating a Second Life account, downloading and installing the Second Life viewer, and having a microphone and headphones available. Those interested were provided with a PDF file with a practical guide to using Second Life.

### 2.3. Evaluation and Data Analysis

At the end of the meeting, the participants were asked to complete a questionnaire (see Appendix A) made up of statements to be answered on a Likert scale from 1 (totally disagree) to 5 (totally agree): 16 of them about general aspects of Second Life or the meeting, and 30 asking about each daily session the following statements:The contents were interesting.The extension of the contents was adequate.I was able to follow the presentation with ease.

Afterward, they were asked to rate ten aspects of the experience from 1 to 10 points, and finally, a space was provided to add “anything else” open-ended comments. This questionnaire was used in other published studies on teaching experiences with medical students [19,21,35,36,37] and included minimal modifications to adapt the statements to the present study. The questionnaire showed previously good reliability for the statements related to the teaching activity and the overall evaluation of the project (Cronbach’s alpha ≥ 0.84) and acceptable reliability for the statements about the attendees’ experience in Second Life (Cronbach’s alpha ≥ 0.70) [36,37].

Data were organized in Excel 2019 sheets (Microsoft, Redmond, WA, USA), and the statistical package SPSS v24 (IBM Corporation, Armonk, NY, USA) was used for descriptive statistics. Numeric responses to the perception questionnaire are presented in this manuscript in terms of mean ± standard deviation. Theme analysis using systematic collaborative coding [38] was used to analyze open comments. The first coding was carried out independently by two of the authors, and later the codes were refined and assigned definitively in group consensus meetings, in which the identification of themes and their definitions was carried out.

## 3. Results

### 3.1. Participation

Fifty-eight radiology residents (40 women and 18 men) took part in one of the four clinical session meetings. Thirty of them (52%) did not start the activity or abandoned it after attending for one or two days. The 28 (48%) who attended the meetings (20 women and 8 men) were made up of 13 first-year, 9 second-year, 2 third-year, and 3 fourth-year residents, trained in 21 different hospitals located in 18 Spanish cities: Málaga (3), Seville (3), Móstoles (3), Madrid (2), Córdoba (2), Marbella (2), Barcelona (2), Cádiz, Bilbao, Granada, Salamanca, Cáceres, Teruel, Mérida, Getafe, Majadahonda, Las Palmas de Gran Canaria, and Oviedo. Attendance was checked by asking the participants to deliver a notecard to the organizers. Notecards are in-world messages that are sent from one avatar to another, recording the avatar that created it, the date and time of creation and delivery. The participants attended a mean of 6.8 ± 1.9 days, with a median of 7 days. Twenty-two of them (78.6%) attended six or more daily sessions.

### 3.2. Evaluation of the Experience

#### 3.2.1. Quantitative Assessment

Twenty-three participants (81.2%) completed and handed in the evaluation questionnaire. Between 95.7% and 100% of those surveyed agreed (Likert values 4–5) that the initiative was interesting, the island’s environment attractive, the postgraduate building suitable for holding the meeting, the content appropriate to their training as residents, the teachers’ intervention adequate, and the programming of the meeting interesting, with mean values of 4.74–4.96 (Figure 3). Most of the residents had no prior knowledge of Second Life (1.91 ± 1.47). Only four (17.4%) agreed to know this virtual platform beforehand. The aspects with the lowest agreement were the ease of moving through Second Life (3.22 ± 0.80) and the ease of creating and managing their avatar (3.83 ± 0.8). Overall, the computer requirements and the internet connection were adequate to run Second Life (4.13 ± 1.10 and 4.57 ± 0.79, respectively). Only four respondents agreed that the contents were difficult, while 18 (78.3%) disagreed with this statement. Seventeen (73.9%) agreed that their participation in the meeting was very active, and 19 (82.6%) agreed that contact with their peers was important for their training. Twenty-one respondents (95.7%) agreed they were willing to repeat a Second Life experience this year, and 100% agreed they were willing for the following year.

Regarding the questions about all the daily sessions, 98.6% of the respondents agreed that the content was interesting, 94.6% agreed that the length of the content was adequate, and 94.9% agreed that they could easily follow the presentations. All responses to the three statements yielded mean Likert values of 4.79 ± 0.44, 4.65 ± 0.60, and 4.70 ± 0.58, respectively. Figure 4 shows these values broken down for each daily session.

The mean rating from 1 to 10 points was higher or equal to 8.9 (Figure 5), which means a very positive perception of the overall experience, the organization of the project, the 3D environment, the contents, the interaction with colleagues and the work done by participants. The best rated of all the items included were the teachers (9.7 ± 0.6) and the usefulness of their training (9.4 ± 0.9).

#### 3.2.2. Qualitative Assessment

Nineteen of the 23 questionnaires delivered (82.6%) contained open comments (these are included verbatim, translated into English, in Appendix A). After the thematic coding, 10 different codes were identified. After code analysis, themes were generated and named (Table 2). All comments contained more than one topic and were, therefore, coded with different codes. On 12 occasions, users provided **positive** comments about the experience, with various expressions such as “delighted to have participated”, “interesting”, “fabulous”, “good”, “complete and enriching”, “spectacular and highly motivating”, or directly “positive”. There were 12 comments expressing favorable opinions about the **Second Life** platform, which were sub-coded in a second layer. Four comments spoke about the ubiquity of contact with other colleagues from different hospitals, three comments highlighted the immersive aspect of the platform, three found it interesting, and two comments highlighted its educational potential. These comments included some very interesting and detailed descriptions, such as the following:


*“The environment had me hooked, and especially the issue of doing it connected from home, but in real time.”*



*“The audio is very immersive (hearing it depending on where the camera is, the sound that fades when the speaker moves away). Also very well achieved the feeling of being in a real classroom, but with the comfort of being at home.”*



*“Undoubtedly, this environment has a potential that can be used in many ways.”*


Eighteen comments provided feedback on the clinical session **meeting**, highlighting valuable aspects of it. They were also sub-coded in a second layer. Nine comments spoke of the fact that it was a useful strategy for the sought teaching objective, to improve public presentations, for example:


*“At no time during our training in hospitals are we taught to give presentations, nor are we given constructive criticism on how to improve them, so I was delighted to be part of this course.”*


Six comments about the meeting were sub-coded because they highlighted the interaction and exchange of opinions with their peers. Two comments highlighted the relaxed and pleasant atmosphere of the meeting, and one comment emphasized the formative assessment provided by the meeting.

Ten comments were coded because they expressed a **willingness** to participate in similar meetings. Seven comments expressed **thanks** for the effort made to hold the meeting, for allowing them to participate, or simply thanked. Four comments addressed the **recommendation** of the meeting to their fellow residents. Four comments described **technical** problems: (1) with connectivity due to the limited capacity of their PC; (2) with the movements and actions of the avatar; (3) with the internet connection, providing interruptions in the audio and slideshow; or (4) with the audio of the recorded presentations. In four comments, the **duration** of the meeting (10 sessions) was found to be excessive, proposing to reduce them. Three comments provided **suggestions** for other radiology courses to be delivered in the virtual world of Second Life. Finally, three comments that talked about the participation of more residents, the personal preference for face-to-face meetings or other educational elements found on the island were coded as **others**.

## 4. Discussion

An immersive experience in the multi-user virtual world of Second Life has been designed for radiology residents, with the aim of training public communication skills by presenting clinical sessions in meetings consisting of 10 two-hour sessions over four weeks. Four editions were carried out, in which the participants found the 3D environment attractive and suitable for learning and the experience interesting and useful, pointing out the formative value for the training of communication skills in public oral presentations.

### 4.1. Multi-User Virtual Environments to Train Public Communication

Second Life offers the potential for new pedagogical approaches in medical education [39], makes it possible to create a realistic environment, provides a platform for veritable communication between users, and even allows “face-to-face” interaction through avatars [17]. Some significant advantages of 3D environments have been described compared to synchronous communication platforms based on 2D technology [15], which represent interesting advantages of the educational experience carried out in this study:*Presence.* They induce a strong sense of presence (sensation of “being there”) in users, who move in a computer-generated virtual space, reacting to actions and changing their point of view on the scene with movement, and producing a strong sense of co-presence when other avatars are present [14]. Some participants in the present study have specifically pointed this out in their open comments. The stronger the feeling of being part of the virtual environment, the more meaningful the experience. Presence and learning are strongly related, as increasing presence also increases learning and performance.*Awareness*. Multi-user virtual environments are characterized by the interaction between remote users. Once in the environment, each user has a first-person perspective in which the actions of the other participants take place. The user participates actively, not just watches. In relation to this, social awareness refers to the ability to sense the presence and location of the co-participants in a learning environment, reinforcing the perception of “Who is there” and “What is happening” [15]. The residents who participated in this study rated the virtual interaction with their peers very highly (8.9 ± 1.5), agreed that contact with them is beneficial for their learning, and highlighted in six open comments the interaction and exchange of experiences in Second Life, even without knowing each other personally, as something very valuable.*Communication.* Verbal messages are exchanged in an underlying emotional context that needs to be understood and communicated. Despite not having gestural communication in Second Life, the voice gives it an important immersive component [15]. It provides very personal characteristics of each one, such as the accent, the timbre or the tone when speaking. The meeting of the clinical sessions was largely based on voice communication, which reinforces the social aspect of communication between the participants.*Belonging to a community.* In order to improve learning, it is important to create a virtual social space designed to favor informal encounters between students as classmates [40] or between residents of the same *medical* specialty, as in this study. By participating in a learning program, people are interested not only in gaining knowledge but also in gaining a positive sense of efficacy, that is, a feeling that they have had some effect in this environment.

Second Life is an ideal tool for the development of transversal oral communication skills. Experiences have been described that demonstrate Second Life’s usefulness in improving communication skills with patients [41]. The participants in this study mostly recognize the clinical session meetings in Second Life as an effective method of training in transversal communication skills, such as the public oral presentation of a topic, interacting with other users in real time, and establishing bonds and social relationships among attendees. As a direct consequence, they all showed a willingness to repeat similar experiences. Communication and presentation skills are essential in the career of a doctor. Feedback comments from the audience, along with expert observation, are elements that help improve these skills [1]. These meetings contribute both elements, through direct verbal interaction in the sessions, especially in the rounds of interventions after the presentations, in which all those present took turns to intervene in a pleasant and relaxed atmosphere. Van Ginkel et al. [42] conducted a study with 36 first-year college students who completed a required oral presentation course in a virtual environment built with Unity (Unity Technologies, San Francisco, CA, USA). We agree with these authors that this type of virtual experience imitating real life helps to further develop oral presentation skills.

The presenter’s voice is literally the instrument of connection with the attendees. When he/she gives a presentation, how the audience feels about his/her voice is integral to how they perceive the presentation [1]. This study demonstrates the potential usefulness of Second Life in improving oral scientific communication skills. The quality of the audio allows interaction, evaluation, and improvement of oral expression. On the one hand, virtual worlds like Second Life have the disadvantage of not perceiving gestural communication, but on the other hand, the filter provided by the interface and avatar in immersive environments makes those who are starting out feel less embarrassed when speaking in public [26].

In this study, users have valued the experience as interesting and appropriate for their training as radiology residents. One of the main advantages lies in the synchronous participation from different locations, saving travel costs for residents and teachers. But our study also brought an experiential approach to attendees. Experiential approaches have proven to be an effective educational initiative to teach important communication skills to radiology trainees who have not learned well using traditional didactic approaches [8]. Unlike traditional distance learning using 2D platforms, Second Life provides a more personal learning experience, which is why many researchers and teachers prefer it. Second Life can also provide a platform for more informal interaction between students and teachers. Traveling to architectural sites, visiting art galleries and science museums, and attending musical performances with an instructor are all possible (and convenient) for Second Life students [17]. Second Life’s informal environment allows trainees to feel comfortable interacting with the instructor and other trainees, which generates a great sense of engagement with the group, as occurred in our study with the visits made at the end of the presentations of daily clinical sessions.

### 4.2. Educational Design of the Clinical Sessions Meetings

When analyzing the educational design of the clinical session meetings carried out in this study within the framework of the seven design principles to develop oral presentation competence proposed by Van Ginkel et al. [43] (p. 68), it can be verified that it complies with these principles:Instruction
1.The learning objectives were explicitly communicated to the participants and specifically formulated in relation to the criteria for the presentations in the clinical sessions.2.The learning and presentation tasks were directly related to the radiology medical specialty, which contributes to improving self-efficacy beliefs, oral presentation competence, and reducing communication apprehension.Learning activities
3.Each presentation provided opportunities for residents to observe peer or expert role models to increase self-efficacy beliefs and oral presentation competence.4.The meeting provided opportunities for residents to practice their oral presentations and develop their oral presentation proficiency by decreasing their communicative apprehension.Evaluation strategy
5.Feedback through a round of interventions was explicit, contextual, and immediate to improve students’ oral presentation proficiency.6.The meeting encouraged peer participation in formative assessment to develop residents’ oral presentation competency and attitudes toward presentation.7.Video recordings of the sessions facilitated self-assessment to encourage residents’ self-efficacy beliefs, oral presentation competence, and attitudes toward presentation.

### 4.3. Limitations

This study has some limitations that must be considered. Firstly, participation was low. Although the current data allow us to have a pilot of how the residents perceived the meeting of clinical sessions, it is necessary to carry out new experiences improved with what has been learned. A larger sample is needed to examine population effects and to compare subgroups (e.g., sex or years of residence). Related to this, there was a high dropout rate after initial enrollment. This may be because it was a free activity divided into 10 sessions over one month. It is often said that “people don’t value things they get for free.” In any case, the fact that it was a free registration makes it easier for those registered to decide not to participate or leave due to scheduling problems. For this reason, a symbolic fee was introduced in other activities with residents in Second Life carried out later. Some attendees considered the length of the meeting excessive, as expressed in four open comments. This could be an impediment to registering or continuing to participate, so for future meetings, the number of days should be reduced, probably to five or six.

Secondly, the use of Second Life has some limitations and drawbacks. Some learning time is required to use the interface and manage properly in the virtual world, but in general, everyone learns enough after basic training. In this study, only four residents (17.4%) did not agree with the correct handling of Second Life, but three of them strongly agreed with following all the sessions fluently. It is essential that attendees learn to correctly use the basic audio communication controls (microphone and headphones) and the vision of the avatar to be able to follow the development of a session properly [26]. There are also technological limitations because reproducing the world needs a minimum computer system and graphics card requirements, as well as enough data transfer over the internet [16]. These tend to affect a few users in the medical setting, between 9% and 11% of the participants [20,21,36]. In this study, only two residents disagreed with having enough computer power, and only one disagreed with having enough network connection to run Second Life properly. All three strongly agreed that they followed the sessions fluently.

Thirdly, the time invested in the creation and management of the meeting is an inconvenience to consider. Designing, validating, and executing teaching activities takes time, and the overhead associated with designing, implementing, and practicing Second Life requires effort and skill on the part of educators [14]. For this reason, it is important to reuse resources and virtual settings in new events, taking advantage of the practice acquired as instructors. Educational activities in radiology carried out at Medical Master Island to date exceed 3,500 undergraduate and postgraduate students.

### 4.4. Future Proposals

It has been proposed that interaction with virtual audiences to promote competence in oral presentations should be further developed in three-dimensional virtual technology [42]. The current experience describes a pilot course, reproducible, with a positive perception of the participants in terms of substance and form. According to this, the authors propose to use multiple virtual worlds, such as Second Life, as a work environment to carry out training courses in public communication competencies. The methodology of these courses, based on synchronous sessions, should allow the attendees to give an oral presentation, followed by a round of interventions of the attendees, providing instant feedback from peers and experts and contributing to the formative assessment of the participants. The proposed structure, in five or six 2 h sessions over three weeks, should be divided into the following: a session of presentation of the course and training of basic functions in Second Life, a teaching session on how to give effective oral presentations, and three to four sessions where participants give their presentations followed by attendees’ intervention rounds. Addressing these sessions to medical residents has the advantage that they usually have previously prepared presentations whose contents are interesting for all attendees. Video recordings of the sessions, with, e.g., open-source software such as OBS Studio v29 (Open Broadcaster Software), can allow participants to reflect on the sessions made or visualize them afterward in the event they cannot attend.

The objective of this study was not to compare the oral presentation of the public in real environments and in virtual worlds, although it would be interesting in the future to carry out a randomized comparative study, such as the one carried out with medical students in seminars on radiography interpretation [21], to demonstrate the differences between the virtual and real environment in this type of courses. A randomized study would be suitable, but the appropriate design would require the economic expense corresponding to travel for residents and teachers from different hospitals to participate in several face-to-face sessions.

Public speaking anxiety is a variant of communication apprehension [22]. Cognitive-behavioral therapy is an effective method of treating communication apprehension, as well as other social phobias [44]. Its foundation is “systematic desensitization” through gradual exposure to the reason that causes apprehension. Virtual environments are effective in triggering real-world reactions to public speaking, and virtual environment-based exposures have been shown to reduce measures of public speaking anxiety [45]. Second Life can be used as an effective tool to allow trainees to practice their presentations before delivering them in a face-to-face meeting and overcome the fear of public speaking. In this study, the level of anxiety when speaking in public of the users was not measured since it was not among the objectives of the study, but it would be very interesting to evaluate it in future studies, for example, through the Public Speaking Anxiety Scale (PSAS) by Bartholomay and Houlihan [46]. In addition, residents’ developments in presentation behavior could be assessed by adopting a rubric instrument, and their attitude towards public presentation could be measured by a self-assessment test, such as those proposed by Van Ginkel et al. [47].

For this type of activity to be successful, based on the presentation of clinical sessions and constructive discussion about them, the topics must be interesting for the participants. This constitutes a fundamental instructional design principle to develop oral presentation competence [43]. In this study, the learning and presentation tasks are directly related to the radiology medical specialty, but the working model developed is easily exportable to other medical specialties simply by changing the content of the clinical sessions.

## 5. Conclusions

This study demonstrates that a meeting for training in public communication skills can be held in an immersive, innovative and playful environment such as Second Life, provides a positive perception of radiology residents on this activity and proposes a structure to hold these meetings.

Second Life can be used effectively to train oral communication skills in public through presentations such as those given in clinical sessions, with the advantage of remote access in an immersive environment with a great feeling of being present. The participants in the clinical session meetings for radiology residents found the three-dimensional environment attractive and suitable for learning and the experience interesting and useful, pointing out the formative value for the training of communication skills in public and oral presentations. They also highlighted the advantages of social contact in virtual interaction with their peers. Oral communication training in virtual worlds should be further explored in other medical specialties and by undergraduate students.

## Figures and Tables

**Figure 1 ijerph-20-04738-f001:**
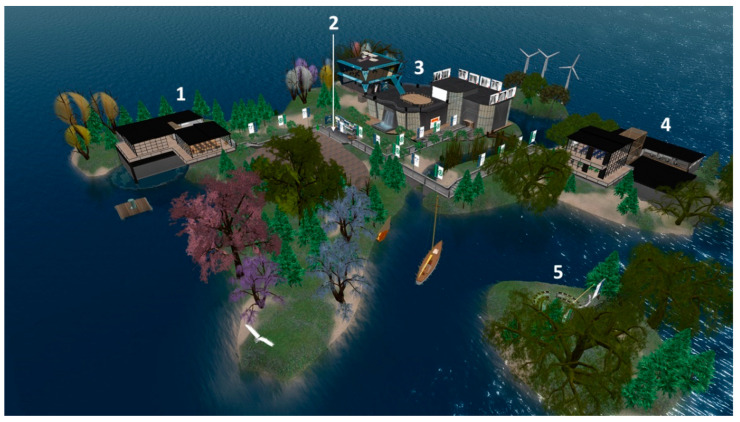
Aerial view of Medical Master Island: (**1**) Postgraduate building, where the present study was carried out; (**2**) point of entry to the islands; (**3**) Medical Master Conference Center; (**4**) Undergraduate building, and (**5**) open-air auditorium on an islet.

**Figure 2 ijerph-20-04738-f002:**
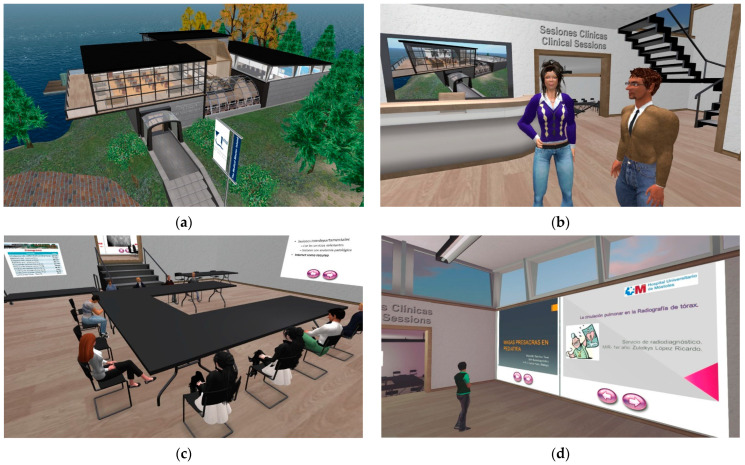
Various images of virtual settings where the present study was carried out: (**a**) exterior view of the Postgraduate Building, (**b**) entrance hall, (**c**) clinical session room with attendees seated around the U-shaped table, and (**d**) exhibition room with the presentations shown on screens wall.

**Figure 3 ijerph-20-04738-f003:**
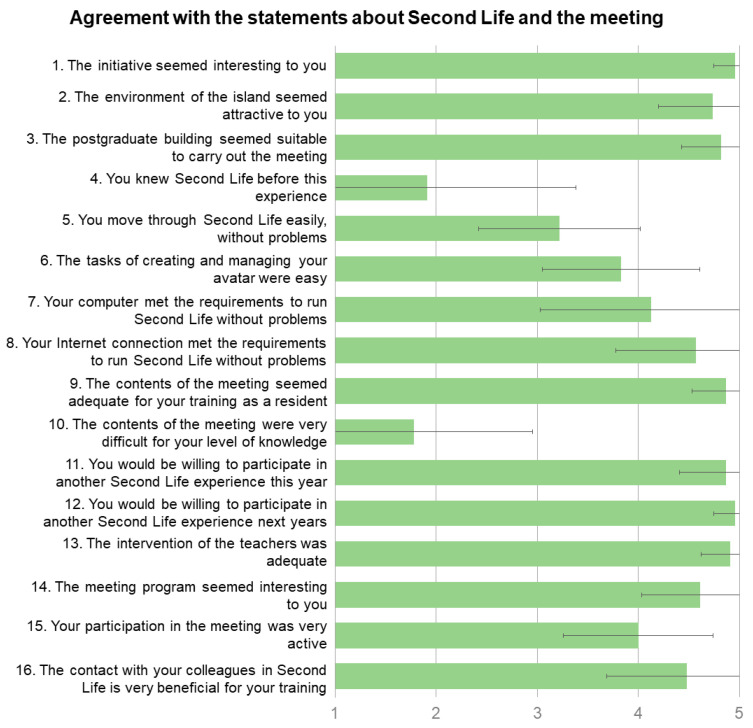
Bar chart with the responses of the participants on a Likert scale from 1 (totally disagree) to 5 (totally agree) related to Second Life and the clinical session meetings. Data are mean values; error bars represent the standard deviation.

**Figure 4 ijerph-20-04738-f004:**
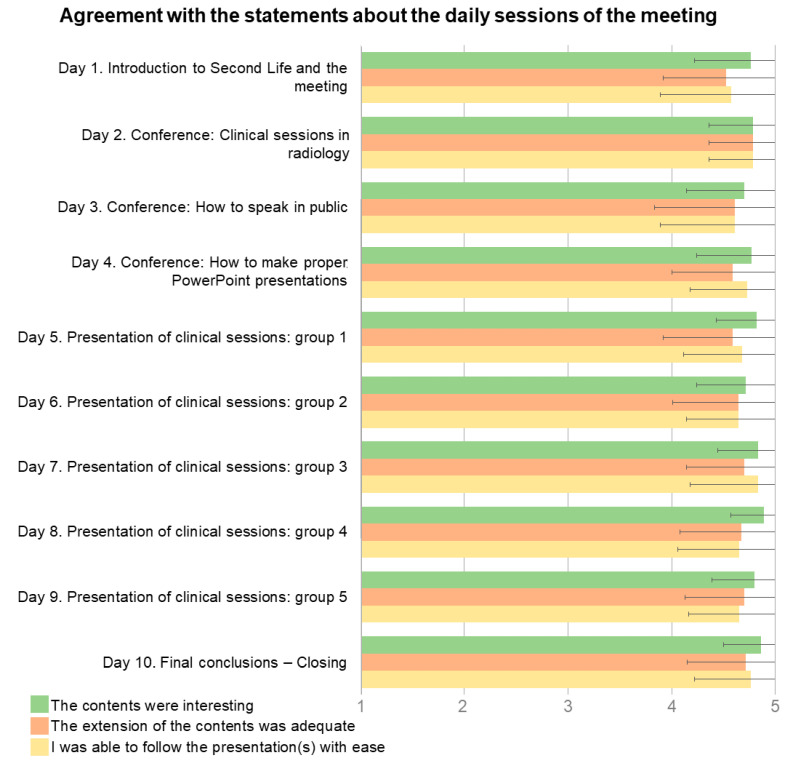
Bar chart with the responses of the participants on a Likert scale from 1 (totally disagree) to 5 (totally agree) related to the daily sessions of the meeting. Data are mean values; error bars represent the standard deviation.

**Figure 5 ijerph-20-04738-f005:**
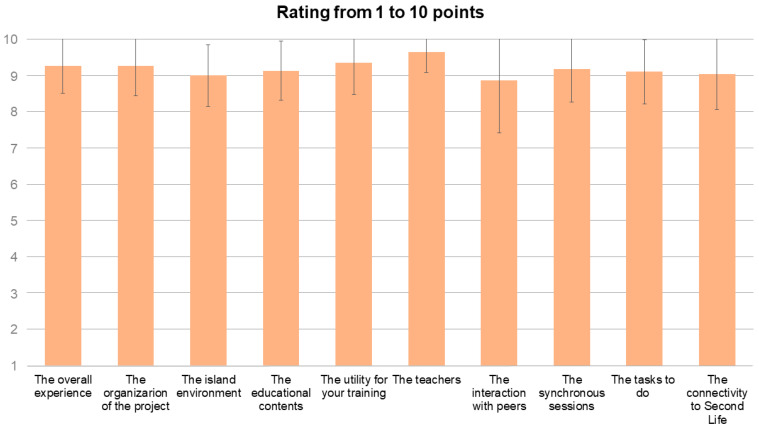
Rating bar graph from 1 to 10 points of various aspects of the experience. Data are mean values; error bars represent the standard deviation.

**Table 1 ijerph-20-04738-t001:** Schedule of the clinical session meetings.

Day	Contents
1	Introduction to Second Life, visit to the island and training with the avatar
2	Conference: Clinical sessions in radiology
3	Conference: How to speak in public
4	Conference: How to make proper PowerPoint presentations
5	Presentation of clinical sessions, group 1
6	Presentation of clinical sessions, group 2
7	Presentation of clinical sessions, group 3
8	Presentation of clinical sessions, group 4
9	Presentation of clinical sessions, group 5
10	Final conclusions—Closing

**Table 2 ijerph-20-04738-t002:** Thematic coding of open comments.

Code	Subcode	Number ^1^	Description of Themes
Positive		12	The overall experience was perceived as positive.
Second Life		12	Feedback on the Second Life platform:
	Ubiquity	4	The ubiquitous contact in real time with colleagues from other hospitals was highlighted.
	Immersive	3	The immersive aspect and the feeling of being in a real classroom were highlighted.
	Interesting	3	The platform was found to be interesting.
	Potential	2	The educational potential of Second Life was highlighted.
Meeting		18	Feedback to the clinical session meeting:
	Useful	9	The meeting was perceived as useful for the teaching objectives.
	Interaction	6	The interaction and exchange of opinions with other peers were highlighted.
	Atmosphere	2	The atmosphere of the meeting was described as relaxed and pleasant.
	Assessment	1	The formative assessment provided by the meeting was emphasized.
Willingness		10	The will to repeat similar experiences was expressed.
Thanks		7	The opportunity to participate in the experience was appreciated.
Recommendation		4	The experience was recommended to other colleagues.
Technical		4	Some technical problems were described.
Duration		4	The duration of the meeting was found excessive, proposing alternatives.
Suggestion		3	Other radiology courses in Second Life were suggested.
Others		3	Other themes not included in the above were presented.

^1^ Number of times the code or subcode was found in the 19 open comments.

## Data Availability

The datasets used and/or analyzed during the current study are available from the corresponding author upon reasonable request.

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
