# Peer review of "Improving Oral Presentation Skills for Radiology Residents through Clinical Session Meetings in the Virtual World Second Life"

_ijerph, 2023, doi:10.3390/ijerph20064738_

Round 1

Reviewer 1 Report

Some suggestions were:

1. Technocially, Second Life is a "old" technology for comon users. VR is a better term, which can be integrated in the current theoretical discussion. I suggested the auhors could find more scientific evidence redgarding virtual world in promoting presentation skills. Some simulation games are good examples. 

2. The quantitative analysis could be further organized. The current format of the quantiative anlaysis is too rough. Bakcound information such as gender could be use to tell the gender effect on the measurement. Also, the authors could use score 4.0 as a benchmark to perform t-test. In such way, you can know which measurement item reach a significant difference.

Author Response

Reviewer 1

Some suggestions were:

  1. Technocially, Second Life is a "old" technology for comon users. VR is a better term, which can be integrated in the current theoretical discussion. I suggested the auhors could find more scientific evidence redgarding virtual world in promoting presentation skills. Some simulation games are good examples. 

R: Thanks for this comment. Second Life is one more platform among those that provide virtual world services, the most used to date in the educational environment. Virtual worlds, unlike the current concept of virtual reality (VR), do not require a headset or special handheld devices, but rather represent the virtual world on a computer monitor. Using the term virtual reality would be imprecise, as this term currently refers to platforms that allow the user to view with head movements with Oculus lens-like devices and interact with hand controls. VR, together with augmented reality and multi-user virtual worlds, make up the different approaches to the well-known current term "metaverse". We have modified the introduction to contextualize this aspect.

The fact that the Second Life platform was developed in 2003 should not lead us to consider it an “old” technology, since it has significantly advanced in the quality of virtual world rendering (especially during the pandemic). With numerous publications in different fields of science and humanities. See, for example, the following search: https://scholar.google.com/scholar?hl=en&as_sdt=0%2C5&as_ylo=2021&q=%22Second+Life%22+virtual&btnG=

  1. The quantitative analysis could be further organized. The current format of the quantiative anlaysis is too rough. Bakcound information such as gender could be use to tell the gender effect on the measurement. Also, the authors could use score 4.0 as a benchmark to perform t-test. In such way, you can know which measurement item reach a significant difference.

R: Thanks for this comment. In this study, descriptive statistics of the quantitative data have been carried out (this is expressly indicated now in the material and method section). Due to the low number of participants and the high proportion of women, it does not make sense to make a comparison between sexes (a comment on this is included in limitations and future trends). Twenty women and 8 men participated in the experience and 18 women, and 5 men answered the questionnaire. Anyway, we have verified the T-test of the students, as you suggest. There are only significant differences between both subgroups (women and men) in two items. In one, all the women answered 5 and all the men answered 5 except one who answered 4, with p=0.028, illustrating the invalidity of such a comparison.

Reviewer 2 Report

Thank you for this interesting article on skill development in VR environment. The multi-user virtual learning space and the context of radiology education highlight the novelty of the current study. However, I think the article can be improved by addressing the following issues: 

1. The design considerations behind the VR environment need to be further elaborated. Yes, the study proved that the VR environment was effective, but what design features made it effective? I think a more systematic and detailed description of instructional design behind the intervention would bring more value to the general readers. 

2. The quantitative results are based on self-reported questionnaire and are mainly descriptive, so I am glad that the authors included qualitative data and themed findings. However, I think the coding scheme and analysis methods need to be better explained. For instance, the section of 3.1.2 ends with a table of codes... I found it baffling since usually we identified the open coding and then extracted the themed findings from contrasting or synthesizing those codes. (I think the numbers of headings are also incorrect in this section). 

3. The discussion needs to be more closely tied to the results. Please use this section to explain and interpret the study results presented in the previous section, and/or compare with existing literature with a discussion of similar or contradictory findings. Do not introduce new findings; Do not describing more empirical evidence; Do not repeat the study results. 

Author Response

Thank you for this interesting article on skill development in VR environment. The multi-user virtual learning space and the context of radiology education highlight the novelty of the current study. However, I think the article can be improved by addressing the following issues: 

R: We thank reviewer 2 for his/her words and opinions about our study. We appreciate the proposed issues, and we will answer them below, we hope satisfactorily.

  1. The design considerations behind the VR environment need to be further elaborated. Yes, the study proved that the VR environment was effective, but what design features made it effective? I think a more systematic and detailed description of instructional design behind the intervention would bring more value to the general readers. 

R: We thank so much reviewer 2 for this comment. A more detailed description of instructional design in material and method has been added, making direct reference to its two components (curriculum design and learning design) in two subsections of material and methods. Following your suggestion, a new subchapter is added in the discussion, where reference is made to the fact that the educational design of this study followed the design principles and advantages to develop oral presentation competence proposed by Van Ginkel et al. 2015.

Van Ginkel, S.; Gulikers, J.; Biemans, H.; Mulder, M. Towards a set of design principles for developing oral presentation competence: A synthesis of research in higher education. Educ Res Review 2015, 14, 62-80. [doi.org/10.1016/j.edurev.2015.02.002]

  1. The quantitative results are based on self-reported questionnaire and are mainly descriptive, so I am glad that the authors included qualitative data and themed findings. However, I think the coding scheme and analysis methods need to be better explained. For instance, the section of 3.1.2 ends with a table of codes... I found it baffling since usually we identified the open coding and then extracted the themed findings from contrasting or synthesizing those codes. (I think the numbers of headings are also incorrect in this section). 

R: We thank reviewer 2 for this comment. In the consensus meetings the meaning of the codes was described, although what was really done was to identify the themes and define them. The material and methods section has been modified accordingly, indicating this. Table 2 has been modified in the results, including the definition of the themes found. These issues are discussed as results of the qualitative analysis at various points in the discussion.

  1. The discussion needs to be more closely tied to the results. Please use this section to explain and interpret the study results presented in the previous section, and/or compare with existing literature with a discussion of similar or contradictory findings. Do not introduce new findings; Do not describing more empirical evidence; Do not repeat the study results. 

R: We thank reviewer 2 for this comment. In response to this and some from reviewer 3, we have modified the discussion substantially, without introducing new findings or more empirical evidence or repeating the results of the study. Some sections have been moved to the introduction to further contextualize the present study. Thank you.

Reviewer 3 Report

The paper is original and interesting in terms of its subject matter, as it uses immersive technology in virtual worlds to carry out simulated activities instead of those same activities in the real world, something that is novel and trendy in today's society. However, the case study conducted has a very small sample size and the statistics derived from the data collected are not clear that they can be taken as valid with such a small sample. For example, in order to validate the Chronbach's alpha values, a larger sample would normally be necessary (in the case study an effective sample of N=23 is available), taking into account the number of items being analysed. However, to confirm this, a PCA analysis (Bujang, Omar, & Baharum, 2018; Yurdugül, 2008) would have to be performed first, although a larger sample of 30 would normally be necessary.

The paper is easy to read, although there are parts that should be revised in terms of wording and English language syntax, e.g., in the first paragraph of section 2.1. In addition, it is redundant on many occasions throughout the paper.

The main concern is that only conclusions or findings that the virtual environment used is interesting and useful are drawn from the data extracted from the qualitative analysis, without comparing it with a non-virtual environment as a control group. In summary, the concerns found are:

- In the introduction there is a lack of critical work and analysis on other similar studies (it is mentioned that they exist, but they are not analysed).

- The statistical analysis is not rigorous.

- There is no control group to compare the virtual experience with a non-virtual experience, not being able to confirm whether the course would have been as successful and interesting in a non-virtual experience, or whether the difference would have been significant.

- It does not add to the existing literature any findings beyond the fact that the environment is interesting and useful to participants.

- It is not clear why it is particularly suitable for radiology residents rather than other groups who could also benefit from a course in effective communication, which is needed in many other productive areas. For example, in section 4, it is specified that the course has been designed specifically for radiology residents but it is not clear why this course is different from another course also on communication skills but for another group.

- No methodology or protocol is proposed for conducting this type of activity in virtual environments that could be derived from the study and that could enrich the literature, beyond the determination to reduce the specific course to 5 or 6 days in the future (section 4.4).

- The questionnaires do not collect data that could have been subsequently analysed to draw interesting conclusions, for example, in section 4.2, public speaking anxiety is mentioned, but the participants are not asked whether they suffer from this type of anxiety initially and whether they think it has improved at the end of the course, so no conclusions can be drawn in this respect, nor can these issues be discussed in the paper. At the end of this section, it is stated that the use of Second Life improves this apprehension about public speaking, but the truth is that no data is collected in this case study, so it would be more appropriate for this section to be in the introduction, because this statement is not substantiated. 

- In section 4.3, again the claims in this case study derived from the data analysis are not verified.  The advantages of this environment cannot be confirmed without comparison with a group using a non-virtual environment, for example when it is said that it reduces the embarrassment of public speaking, this is not proven by data.

- In section 4.4, it is mentioned that only 4 residents (17.4% of the sample) could not handle themselves well in the environment, but this percentage is relatively large compared to the case study sample.

Bujang, M.A., Omar, E.D., & Baharum, N.A. (2018). A review on sample size determination for Cronbach's alpha test: a simple guide for researchers. The Malaysian journal of medical sciences: MJMS, 25(6), 85.

Yurdugül, H. (2008). Minimum sample size for Cronbach's coefficient alpha: A Monte-Carlo study. Hacettepe Üniversitesi eğitim fakültesi dergisi, 35(35), 1-9.

Author Response

The paper is original and interesting in terms of its subject matter, as it uses immersive technology in virtual worlds to carry out simulated activities instead of those same activities in the real world, something that is novel and trendy in today's society. However, the case study conducted has a very small sample size and the statistics derived from the data collected are not clear that they can be taken as valid with such a small sample. For example, in order to validate the Chronbach's alpha values, a larger sample would normally be necessary (in the case study an effective sample of N=23 is available), taking into account the number of items being analysed. However, to confirm this, a PCA analysis (Bujang, Omar, & Baharum, 2018; Yurdugül, 2008) would have to be performed first, although a larger sample of 30 would normally be necessary.

R: We sincerely appreciate the reviewer 3's words about our study. In fact, the number of residents who participated in this study is low. This is considered the first limitation of this study in the current version, and so is now expressed in the corresponding section. The current experience describes a reproducible pilot course with a positive perception of students and teachers. We believe that it is necessary to carry out new experiences improved with what has been learned. A model based on the current one is proposed. The questionnaire was used in other published studies on teaching experiences with medical students [19,21,35-37] and includes minimal modifications to adapt the statements to the present study. The questionnaire showed previously good reliability for the statements related to the teaching activity and the overall evaluation of the project (Cronbach's alpha ≥0.84) and acceptable reliability for the statements about the attendees' experience in Second Life (Cronbach's alpha ≥ 0.70) [36,37].

The paper is easy to read, although there are parts that should be revised in terms of wording and English language syntax, e.g., in the first paragraph of section 2.1. In addition, it is redundant on many occasions throughout the paper.

R: We thanks reviewer 3 for this comment. The wording of the entire text has been revised again, avoiding excessively long sentences and reorganizing them, as in the first paragraph of section 2.1.

The main concern is that only conclusions or findings that the virtual environment used is interesting and useful are drawn from the data extracted from the qualitative analysis, without comparing it with a non-virtual environment as a control group. In summary, the concerns found are:

R: We appreciate this comment, but we must make it clear to the reviewer that this is not a case control study designed as such. This study aimed to design and conduct a meeting of clinical sessions in the Second Life virtual world to improve the oral presentation skills of radiology residents, and to assess the perception of the attendees. It has not been the object of this study to compare the real world with the virtual environment. In the opinion of the authors, nothing can improve real contact between humans, but when distance, for example, is an obstacle, online solutions offer interesting communication possibilities. 3D virtual worlds offer online training resources that allow attendees a great sense of presence, awareness of what is happening there, means of remote communication and a sense of belonging to a community, in a much more palpable way than in 2D environments (these differences have been pointed out by other authors and are expressed in the discussion). Now we have included in the section " future proposals" that the objective of this study was not to compare the oral presentation of the public in real environments and in virtual worlds, although it would be interesting in the future to carry out a randomized comparative study, such as the one carried out with medical students in seminars on radiography interpretation [Lorenzo-Alvarez et. al., 2019]

[Lorenzo‐Alvarez, R.; Tudolphi‐Solero, T.; Ruiz‐Gomez, M.J.; Sendra‐Portero, F. Medical student education for abdominal radiographs in a 3D virtual classroom versus traditional classroom: A randomized controlled trial. AJR Am. J. Roentgenol. 2019, 213,644–650. https://doi.org/10.2214/AJR.19.21131].

- In the introduction there is a lack of critical work and analysis on other similar studies (it is mentioned that they exist, but they are not analysed).

R: Reviewer 3 is right; we appreciate this comment. In the introduction we have included more reference to experiences about the possibilities of Second Life as a tool for synchronous training in radiology, as well as training in public communication skills, concluding with the following paragraph prior to the hypothesis and objectives: "Residents of Radiology, like those in other specialties, would benefit from specific courses that allow them to hone their public communication skills. Multi-user virtual world platforms like Second Life offer interesting cost-effective resources to explore."

- The statistical analysis is not rigorous.

R: We thanks reviewer 3 for this comment. With all due respect, the statistics of quantitative analysis are descriptive. This is best stated in the text at present. As detailed in previous responses, there was no control group, and the number of participants does not allow for subclass comparison.

- There is no control group to compare the virtual experience with a non-virtual experience, not being able to confirm whether the course would have been as successful and interesting in a non-virtual experience, or whether the difference would have been significant.

R: We thanks reviewer 3 for this comment, please read the answer about it above.

- It does not add to the existing literature any findings beyond the fact that the environment is interesting and useful to participants.

R: We thanks reviewer 3 for this comment. With all due respect, we consider this study add to the existing literature. To our knowledge, no training course in public speaking skills in a clinical context, aimed at residents of a medical specialty, has been experimented in multi-user virtual worlds. The description of its characteristics and the work model proposal is a contribution. The positive perception of users and the nuances they provide are essential as the first step in validating the experience. Online social contact, especially in virtual worlds, has interesting characteristics for training communication skills. The model is repeatable and exportable to other medical specialties.

- It is not clear why it is particularly suitable for radiology residents rather than other groups who could also benefit from a course in effective communication, which is needed in many other productive areas. For example, in section 4, it is specified that the course has been designed specifically for radiology residents but it is not clear why this course is different from another course also on communication skills but for another group.

R: We thank reviewer 3 for this comment. The decision to carry out this communication skills course only with radiology residents is because we are dedicated to educational innovation in radiology. But sure enough, it could be extended to many other different medical specialties, as stated at the end in the limitations and future proposals section: "For this type of activity, based on the presentation of clinical sessions and constructive discussion about them, to be successful, the topic must be interesting to the participants. This constitutes a fundamental instructional design principle to develop oral presentation competence [VanGinkel-2015]. In this study, the learning and presentation tasks are directly related to the medical specialty of radiology, but the work model developed is easily exportable to other medical specialties, simply by changing the content of the clinical sessions."

- No methodology or protocol is proposed for conducting this type of activity in virtual environments that could be derived from the study and that could enrich the literature, beyond the determination to reduce the specific course to 5 or 6 days in the future (section 4.4).

R: The reviewer is right, and it was a major omission. Thank you for the observation. This study demonstrates that a meeting for training in public communication skills can be held in an immersive, innovative and playful environment such as Second Life, provides the perception of radiology residents on this activity, and proposes a structure to hold these meetings. This has been included as a startin point in the conclusions section. The new “future proposals” section includes a protocol proposal derived from this study, to be considered by the reader, as well as new assessment possibilities.

- The questionnaires do not collect data that could have been subsequently analysed to draw interesting conclusions, for example, in section 4.2, public speaking anxiety is mentioned, but the participants are not asked whether they suffer from this type of anxiety initially and whether they think it has improved at the end of the course, so no conclusions can be drawn in this respect, nor can these issues be discussed in the paper. At the end of this section, it is stated that the use of Second Life improves this apprehension about public speaking, but the truth is that no data is collected in this case study, so it would be more appropriate for this section to be in the introduction, because this statement is not substantiated. 

R: We thank reviewer 3 for this comment. We have redone the discussion, leading to more solid proposals and conclusions. The old sections "4.1. Medical education in public speaking skills" and "4.2. Public speaking anxiety" have been moved to the introduction, as suggested. Indeed, in this study the level of anxiety when speaking in public of the users has not been measured, since it was not among the objectives of the study, but it would be very interesting to evaluate it in future studies, for example through the Anxiety Scale. of Public Speaking (PSAS) by Bartholomay and Houlihan [Bartholomay-2016]. This is provided in the future proposals of this work, which are in development for this year 2023.

- In section 4.3, again the claims in this case study derived from the data analysis are not verified.  The advantages of this environment cannot be confirmed without comparison with a group using a non-virtual environment, for example when it is said that it reduces the embarrassment of public speaking, this is not proven by data.

R: We thanks reviewer 3 for this comment. With all due respect, we disagree with reviewer 3 on this point. The advantages of this platform can be included in the discussion based on the work published by others and contextualize our results. The former section 4.3, now 4.1, includes, in each paragraph, results of our experience related to conclusions and affirmations of previous publications on Second Life, its particularities and its applicability for the development of oral communication skills. As we have said before, this is not a comparative study and, in our opinion, virtual worlds do not replace face-to-face training, except when this is not possible (for example, when the displacements of the participants do not make the course cost-efficient). We have substantially modified the discussion to clarify these and other issues.

- In section 4.4, it is mentioned that only 4 residents (17.4% of the sample) could not handle themselves well in the environment, but this percentage is relatively large compared to the case study sample.

R: We thank reviewer 3 for this comment. The percentage is relatively large given the small sample size and this is considered a limitation. In fact, only 4 residents (17.4%) did not agree with the correct handling of Second Life, but three of them strongly agreed with following all the sessions fluently. It is essential that attendees learn how to correctly use the basic audio communication controls (microphone and headphones) and the vision of the avatar to properly follow the development of a session. These clarifications are included in the limitations section.

Round 2

Reviewer 1 Report

The authors already provide solutions to the problems identified in the previous review process. The current format of the manuscript is ready for publication.

Reviewer 2 Report

I thank the authors for their revision. I think the quality of the paper has improved as a result. 

Reviewer 3 Report

The paper still has some basic flaws in terms of the case study, but it can somewhat enrich the state of the art.